# Variational Delayed Policy Optimization

**Qingyuan Wu** *
University of Southampton

**Simon Sinong Zhan** *
Northwestern University

**Yixuan Wang**
Northwestern University

**Yuhui Wang**
King Abdullah University of Science and Technology

**Chung-Wei Lin**
National Taiwan University

**Chen Lv**
Nanyang Technological University

**Qi Zhu**
Northwestern University

**Chao Huang** †
University of Southampton

## Abstract

In environments with delayed observation, state augmentation by including actions within the delay window is adopted to retrieve Markovian property to enable reinforcement learning (RL). However, state-of-the-art (SOTA) RL techniques with Temporal-Difference (TD) learning frameworks often suffer from learning inefficiency, due to the significant expansion of the augmented state space with the delay. To improve learning efficiency without sacrificing performance, this work introduces a novel framework called Variational Delayed Policy Optimization (VDPO), which reformulates delayed RL as a variational inference problem. This problem is further modelled as a two-step iterative optimization problem, where the first step is TD learning in the delay-free environment with a small state space, and the second step is behaviour cloning which can be addressed much more efficiently than TD learning. We not only provide a theoretical analysis of VDPO in terms of sample complexity and performance, but also empirically demonstrate that VDPO can achieve consistent performance with SOTA methods, with a significant enhancement of sample efficiency (approximately 50% less amount of samples) in the MuJoCo benchmark. Code is available at https://github.com/QingyuanWuNothing/VDPO.

## 1 Introduction

Reinforcement learning (RL) has achieved considerable success across various domains, including board game [32], video game [27], cyber-physical systems [40, 41, 43]. Most of these achievements lack stringent timing constraints, and, therefore, overlook delays in agent-environment interaction. However, delays are prevalent in many real-world applications stemming from various sensors, computation, etc, and significantly affect learning efficiency [17], performance [6], and safety [26]. While observation-delay, action-delay, and reward-delay [10] are all crucial, observation-delay receives the most attention [7, 33, 42]. Unlike reward-delay, observation-delay, which is proved to be a superset of action-delay [19, 29], disrupts the Markovian property inherent to the environments. In this work, we focus on the reinforcement learning with a constant observation-delay $\Delta$: at any time step $t$, the agent can only observe the state $s_{t-\Delta}$, without access to states from $t - \Delta + 1$ to $t$.

---

*Equal Contribution
†Correspondence to: Chao Huang, `chao.huang@soton.ac.uk`

38th Conference on Neural Information Processing Systems (NeurIPS 2024).

Augmentation-based approach is one of the promising methodologies [4, 19]. It retrieves the Markovian property by augmenting the state along with the actions within the window of delays to a new state $x_t$, i.e., $x_t = \{s_{t-\Delta}, a_{t-\Delta}, \cdots, a_{t-1}\}$, yielding a delayed MDP. However, the underlying sample complexity issue remains a central challenge. Pioneering works [5, 29] directly conduct classical temporal-difference (TD) learning methods, e.g., Deep Q Network [28] and Soft Actor-Critic [13], over the delayed MDP. However, due to the significant growth of the dimensionality, the sample complexity of these techniques increases tremendously. State-of-the-art (SOTA) methods [20, 39, 42] mitigate this issue by introducing an auxiliary delayed task with shorter delays to help learning the original longer delayed task (e.g., improving the long-delayed policy based on the short-delayed value function). However, sample inefficiency is not addressed sufficiently due to the TD learning paradigm still being affected significantly by the increased delays. The memory-less approach [33] improves the learning efficiency by ignoring the absence of the Markovian property of observation-delay RL and learning over the original state space with a cost of serious performance drop. Therefore, the critical challenge still remains: how to **improve learning efficiency without compromising performance** in the delayed setting.

To overcome such a challenge, we propose Variational Delayed Policy Optimization (VDPO), a novel delayed RL framework. Inspired by existing variational RL methods [1, 2, 25], VDPO can utilize extensive optimization tools to resolve the sample complexity issue effectively via formulating the delayed RL problem as a variational inference problem. Specifically, VDPO operates alternatively: (1) learning a reference policy over the delay-free MDP via TD learning and (2) imitating the behaviour of the learned reference policy over the delayed MDP via behaviour cloning. In the high dimensional delayed MDP, VDPO replaces the TD learning paradigm with the behaviour cloning paradigm, which considerably reduces the sample complexity. Furthermore, we demonstrate that VDPO not only effectively improves the sample complexity, but also achieves consistent theoretical performance with SOTAs. Empirical results show that compared to the SOTA approach [42], our VDPO has significant improvement in sample efficiency (*approximately 50% less amount of samples*) along with comparable performance at most MuJoCo benchmarks.

This paper first introduces notations related to delayed RL and variational RL (Sec. 2). In Sec. 3, we present how to formulate the delayed RL problem as a variational inference problem followed by our approach VDPO. Through theoretical analysis, we show that VDPO can effectively reduce the sample complexity without degrading the performance in Sec. 3.2. Practical implementation of VDPO is presented in Sec. 3.3. In Sec. 4, the experimental results over various MuJoCo benchmarks under diverse delay settings validate our theoretical observations. Overall, our contributions are summarized as follows:

- We propose Variational Delayed Policy Optimization (VDPO), a novel framework of delayed RL algorithms emerging from the perspective of variational RL.

- We demonstrate that VDPO enhances sample efficiency, by minimizing the KL divergence between the reference delay-free policy and delayed policy in a behaviour cloning fashion.

- We illustrate that VDPO shares the same theoretical performance as SOTA techniques, by showing that VDPO converges to the same fixed point.

- We empirically show that VDPO not only exhibit superior sample efficiency but also achieves competitive performance comparable to SOTAs across various MuJoCo benchmarks.

## 2  Preliminaries

**MDP.**   A delay-free RL problem can be formalized as a Markov Decision Process (MDP), denoted by a tuple $\langle \mathcal{S}, \mathcal{A}, \mathcal{P}, \mathcal{R}, \gamma, \rho \rangle$, where $\mathcal{S}, \mathcal{A}$ represent state space and action space respectively, $\mathcal{P} : \mathcal{S} \times \mathcal{A} \times \mathcal{S} \to [0, 1]$ represents the transition function; the reward function is denoted as $\mathcal{R} : \mathcal{S} \times \mathcal{A} \to \mathbb{R}$; $\gamma \in (0, 1)$ is the discount factor, and $p(s_0)$ is the initial state distribution. At each time step $t$, the agent takes the action $a_t \sim \pi(\cdot|s_t)$ based on the current observed state $s_t$ and the policy $\pi : \mathcal{S} \times \mathcal{A} \to [0, 1]$, and then observes the next state $s_{t+1} \sim \mathcal{P}(\cdot|s_t, a_t)$ and a reward signal $r_t = \mathcal{R}(s_t, a_t)$. The objective of an RL problem is to find the policy $\pi$ which can maximize the expected return $\mathbb{E}_{\tau \sim p_\pi(\tau)}[\mathcal{J}(\tau)] := \mathbb{E}_{\tau \sim p_\pi(\tau)}[\sum_{t=0}^{\infty} \gamma^t \mathcal{R}(s_t, a_t)]$ where $p_\pi(\tau)$ is the trajectory distribution induced by policy $\pi$. We use $d^\pi(s_t)$ to denote the visited state distribution of policy $\pi$.

**Delayed MDP.** A delayed RL problem with a constant delay is originally not an MDP, but can be reformulated as a delayed MDP with Markov property based on the augmentation approaches [4, 19]. Assuming the constant delay being $\Delta$, the delayed MDP is denoted as a tuple $\langle \mathcal{X}, \mathcal{A}, \mathcal{P}_\Delta, \mathcal{R}_\Delta, \gamma, \rho_\Delta \rangle$, where the augmented state space is defined as $\mathcal{X} := \mathcal{S} \times \mathcal{A}^\Delta$ (e.g., an augmented state $x_t = \{s_{t-\Delta}, a_{t-\Delta}, \cdots, a_{t-1}\} \in \mathcal{X}$), $\mathcal{A}$ is the action space, the delayed transition function is defined as $\mathcal{P}_\Delta(x_{t+1}|x_t, a_t) := \mathcal{P}(s_{t-\Delta+1}|s_{t-\Delta}, a_{t-\Delta}) \delta_{a_t}(a_t') \prod_{i=1}^{\Delta-1} \delta_{a_{t-i}}(a_{t-i}')$ where $\delta$ is the Dirac distribution, the delayed reward function is defined as $\mathcal{R}_\Delta(x_t, a_t) := \mathbb{E}_{s_t \sim b(\cdot|x_t)}[\mathcal{R}(s_t, a_t)]$ where $b$ is the belief function defined as $b(s_t|x_t) := \int_{\mathcal{S}^\Delta} \prod_{i=0}^{\Delta-1} \mathcal{P}(s_{t-\Delta+i+1}|s_{t-\Delta+i}, a_{t-\Delta+i}) \mathrm{d}s_{t-\Delta+i+1}$, the initial augmented state distribution is defined as $\rho_\Delta = \rho \prod_{i=1}^{\Delta} \delta_{a_{-i}}$.

**Variational RL.** Formulating the RL problem as a probabilistic inference problem [21] allows us to use extensive optimization tools in solving the RL problem. From the existing variational RL literature [30, 36], we usually define $O = 1$ as the optimality of the task (e.g., the trajectory $\tau$ obtains the maximum return). Then the probability of trajectory optimality can be represented as $p(O = 1|\tau)$. Then, the objective of variational RL becomes finding policy $\pi$ with highest log evidence: $\max_\pi \log p_\pi(O = 1)$. Then, we can derive the lower bound of $\log p_\pi(O = 1)$ by introducing a prior knowledge of trajectory distribution $q(\tau)$.

$$\log p_\pi(O = 1) \geq \mathbb{E}_{\tau \sim q(\tau)}[\log p(O = 1|\tau)] - \mathrm{KL}(q(\tau)||p_\pi(\tau)) = \mathrm{ELBO}(\pi, q), \quad (1)$$

where KL is the Kullback-Leibler (KL) divergence and $\mathrm{ELBO}(\pi, q)$ is the evidence lower bound (ELBO) [2, 30]. The objective of variational RL is maximizing the ELBO, which can be achieved by various optimization techniques [1, 2, 9, 30].

## 3 Our Approach: Variational Delayed Policy Optimization

In this section, we present a new delayed RL approach, Variational Delayed Policy Optimization (VDPO) from the perspective of variational inference. By viewing the delayed RL problem as a variational inference problem, VDPO can utilize extensive optimization tools to address sample complexity and performance issues properly. We first illustrate how to formulate delayed RL as the probabilistic inference problem with an elaborated optimization objective. Subsequently, we theoretically show that the inference problem is equivalent to a two-step iterative optimization problem. Then, we present the framework of VDPO along with the practical implementation.

### 3.1 Delayed RL as Variational Inference

Delayed RL can be treated as an inference problem: given the desired goal $O$, and starting from a prior distribution over trajectory $\tau$, the objective is to estimate a posterior distribution over $\tau$ consistent with $O$. The posterior can be formulated by a Boltzman like distribution $p(O = 1|\tau) \propto \exp\left(\frac{\mathcal{J}(\tau)}{\alpha}\right)$ [2, 31] where $\alpha$ is the temperature factor. Based on the above definition, the optimization objective of delayed RL can be defined as follows.

$$\max_{\pi_\Delta} \log p_{\pi_\Delta}(O = 1) = \max_{\pi_\Delta} \log \int p(O = 1|\tau) p_{\pi_\Delta}(\tau) \mathrm{d}\tau, \quad (2)$$

where $p_{\pi_\Delta}(O = 1)$ is the probability of the optimality of the delayed policy $\pi_\Delta$, and $p_{\pi_\Delta}(\tau)$ is the trajectory distribution induced by $\pi_\Delta$. Based on Eq. (1) and Eq. (2), we can also show that the ELBO for optimization purpose is as follows (derivation of Eq. (3) can be found in Appendix B).

$$\log p_{\pi_\Delta}(O = 1) \geq \underbrace{\mathbb{E}_{\tau \sim p_\pi(\tau)}[\log p(O = 1|\tau)]}_{A\uparrow} - \underbrace{\mathrm{KL}(p_\pi(\tau)||p_{\pi_\Delta}(\tau))}_{B\downarrow} = \mathrm{ELBO}(\pi, \pi_\Delta), \quad (3)$$

where $p_\pi(\tau)$ is the trajectory distribution induced by an newly-introduced *reference policy* $\pi$. As shown in Eq. (3), we transform the original optimization problem as a two-step iterative optimization problem: maximizing term $A$ while minimizing term $B$. Next, we detail how our VDPO optimizes objectives $A$ and $B$ separately.

### 3.1.1 Maximizing the performance of reference policy by TD Learning

In this section, we discuss the treatment of term $A$ in Eq. (3) and investigate the performance and sample complexity of reference policy $\pi$ under different MDP settings. Maximizing term $A$ in Eq. (3) is equivalent to maximizing the performance of $\pi$ as follows.

$$\max_\pi \mathbb{E}_{\tau \sim p_\pi(\tau)} \left[ \log p(O = 1|\tau) \right] = \max_\pi \mathbb{E}_{\tau \sim p_\pi(\tau)} \left[ \mathcal{J}(\tau) \right]. \tag{4}$$

For Eq. (4), we can train the reference policy $\pi$ in various MDPs with different delays or even delay-free settings. We show that the performance (Lem. 3.1) and sample complexity (Lem. 3.2) of reference policy $\pi$ are correlated to the specific MDP setting. Based on existing literature [12, 22] and motivated by existing works [42], VDPO chooses training the reference policy in the delay-free MDP for gaining the edge in terms of **performance** and **sample complexity**.

**Performance:** Lem. 3.1 indicates that the performance of the optimal policy is likely decreased by increasing delays. This motivates us to learn the reference policy in the delay-free MDP for proper performance.

**Lemma 3.1** (Performance in delayed MDP, Theorem 4.3.1 in [22]). *Let $\mathcal{M}_1, \mathcal{M}_2$ be two constant delayed MDPs with respective delays $\Delta_1, \Delta_2 (\Delta_1 < \Delta_2)$. For the optimal policies in $\mathcal{M}_1, \mathcal{M}_2$, we have $\mathcal{J}_1^* \geq \mathcal{J}_2^*$.*

**Sample Complexity:** Furthermore, for a specific TD-based delayed RL method (e.g., model-based policy iteration), delays also affect its sample efficiency as stated in Lem. 3.2 that stronger delays will lead to much higher sample complexity, resulting in relative learning inefficiency. Therefore, learning the delay-free reference policy makes VDPO superior in sample complexity compared to learning under delay settings.

**Lemma 3.2** (Sample complexity of model-based policy iteration, Theorem 2 in [12]). *Let $\mathcal{M}$ be the constant delayed MDP with delays $\Delta$. Model-based policy iteration finds an $\epsilon$-optimal policy with probability $1 - \sigma$ using sample size $\mathcal{O}\left( \frac{|\mathcal{X}||\mathcal{A}|}{(1-\gamma)^3 \epsilon^2} \ln \frac{1}{\sigma} \right)$, where $|\mathcal{X}| = |\mathcal{S}||\mathcal{A}|^\Delta$.*

Based on the above analysis and inspired by the existing work [23, 42], VDPO adopts a delay-free policy as the reference policy. More rigorous analyses are presented in Sec. 3.2, and we will detail the practical implementation in Sec. 3.3.

### 3.1.2 Minimizing the behaviour difference by Behaviour Cloning

With a fixed reference policy $\pi$, minimizing term $B$ in Eq. (3) can be treated as behaviour cloning at the trajectory level. However, behaviour cloning at the trajectory level is relatively inefficient compared with training at the state level as we have to collect an entire trajectory before training. We next show that we can directly minimize the state-level KL divergence $\mathrm{KL}(\pi(a_t|s_t)||\pi_\Delta(a_t|x_t))$ as presented in Proposition 3.3.

**Proposition 3.3** (State-level KL divergence, proof in Proposition C.1). *For a fixed reference policy $\pi$, the trajectory-level KL divergence can be reformulated to state-level KL divergence as follows.*

$$\mathrm{KL}(p_\pi(\tau)||p_{\pi_\Delta}(\tau)) = \underbrace{\sum_{t=0}^\infty \int d^\pi(s_t)\mathrm{KL}(\pi(a_t|s_t)||\pi_\Delta(a_t|x_t))\mathrm{d}s_t}_{\text{State-level KL divergence}} + Const., \tag{5}$$

*where* $Const. = \mathrm{KL}(\rho(s_0)||\rho_\Delta(x_0))$

$$+ \sum_{t=0}^\infty \int d^\pi(s_t) \int \pi(a_t|s_t)\mathrm{KL}(\mathcal{P}(s_{t+1}|s_t, a_t)||b(s_t|x_t)\mathcal{P}_\Delta(x_{t+1}|x_t, a_t))\mathrm{d}a_t\mathrm{d}s_t.$$

Since transition dynamics, initial state distributions, and reference policy are all fixed at this point, we can minimize the state-level KL divergence instead of the trajectory-level KL divergence for efficient training, and then the optimization objective becomes as follows.

$$\min_{\pi_\Delta} \mathrm{KL}(p_\pi(\tau)||p_{\pi_\Delta}(\tau)) \Rightarrow \min_{\pi_\Delta} \mathrm{KL}(\pi(a_t|s_t)||\pi_\Delta(a_t|x_t)). \tag{6}$$

In this way, VDPO divides the delayed RL problem into two separate optimization problems including Eq. (4) and Eq. (6). How to practically implement VDPO to solve these optimization problems will be presented in Sec. 3.3.

## 3.2 Theoretical Property Analysis

Next, we explain why our VDPO achieves better sample efficiency compared with conventional delayed RL methods, followed by performance analysis of VDPO.

**Sample Complexity Analysis.** In fact, VDPO can use any delay-free RL method to improve the performance of the reference policy (maximizing $A$). Here, we assume that VDPO maximizes $A$ by the model-based policy iteration, and the sample complexity of maximizing $A$ is $\mathcal{O}\left(\frac{|\mathcal{S}||\mathcal{A}|}{(1-\gamma)^3\epsilon^2}\ln\frac{1}{\sigma}\right)$ as described in Lem. 3.2. And minimizing $B$ in VDPO is equivalent to state-level behaviour cloning which has the sample complexity of $\mathcal{O}\left(\frac{|\mathcal{X}|\ln|\mathcal{A}|}{(1-\gamma)^4\epsilon^2}\sigma\right)$ as stated in Lem. 3.4.

**Lemma 3.4** (Sample complexity of behaviour cloning, Theorem 15.3 in [3])**.** *Given the demonstration from the optimal policy, behaviour cloning finds an $\epsilon$-optimal policy with probability $1-\sigma$ using sample size $\mathcal{O}\left(\frac{|\mathcal{X}|\ln|\mathcal{A}|}{(1-\gamma)^4\epsilon^2}\sigma\right)$.*

Based on Lem. 3.2 and Lem. 3.4, we can drive the sample complexity of VDPO (Lem. 3.5).

**Lemma 3.5** (Sample complexity of VDPO, proof in Lem. C.2)**.** *Assumed that maximizing $A$ in Eq. (3) by model-based policy iteration while minimizing $B$ in Eq. (3) by behaviour cloning, VDPO finds an $\epsilon$-optimal policy with probability $1-\sigma$ using sample size*

$$\mathcal{O}\left(\max\left(\frac{|\mathcal{S}||\mathcal{A}|}{(1-\gamma)^3\epsilon^2}\ln\frac{1}{\sigma}, \frac{|\mathcal{X}|\ln|\mathcal{A}|}{(1-\gamma)^4\epsilon^2}\sigma\right)\right).$$

Then, based on Lem. 3.5, we show that our VDPO has better sample complexity than most TD-only methods (e.g., model-based policy iteration [12], Soft Actor-Critic [5, 20, 42]) as follows.

**Proposition 3.6** (Sample complexity comparison, proof in Proposition C.3)**.** *In the delayed MDP, as $\sigma \to 0$, the sample complexity of VDPO (Lem. 3.5) is less or equal to the sample complexity of model-based policy iteration (Lem. 3.2):*

$$\mathcal{O}\left(\max\left(\frac{|\mathcal{S}||\mathcal{A}|}{(1-\gamma)^3\epsilon^2}\ln\frac{1}{\sigma}, \frac{|\mathcal{X}|\ln|\mathcal{A}|}{(1-\gamma)^4\epsilon^2}\sigma\right)\right) \leq \mathcal{O}\left(\frac{|\mathcal{X}||\mathcal{A}|}{(1-\gamma)^3\epsilon^2}\ln\frac{1}{\sigma}\right).$$

Proposition 3.6 tells us that VDPO can reduce the sample complexity effectively, reaching the same performance but requiring fewer samples compared to model-based policy iteration.

**Performance Analysis.** We investigate the convergence of the delayed policy in VDPO (Lem. 3.7) and show that VDPO can also achieve the same performance as existing SOTAs (Proposition 3.8). As mentioned above, Eq. (4) in VDPO can be solved by existing delay-free RL method (e.g., model-based policy iteration) to learn an optimal reference policy $\pi^*$. Then, we can get the convergence of the delayed policy $\pi^*_\Delta$ via Eq. (6).

**Lemma 3.7** (Convergence of delayed policy in VDPO, proof in Lem. C.4)**.** *Let $\pi^*$ be the optimal reference policy which is trained by a delay-free RL algorithm. The delayed policy $\pi_\Delta$ converges to $\pi^*_\Delta$ satisfying that*

$$\pi^*_\Delta(a_t|x_t) = \mathbb{E}_{s_t \sim b(\cdot|x_t)}\left[\pi^*(a_t|s_t)\right], \forall x_t \in \mathcal{X}. \tag{7}$$

Based on Lem. 3.7, we show that the convergence of VDPO is consistent with that of existing SOTA methods (Proposition 3.8).

**Proposition 3.8** (Consistent fixed point, proof in Proposition C.5)**.** *VDPO shares the same fixed point (Eq. (7)) with DIDA [23], BPQL [20] and AD-SAC [42] for the same delayed MDP.*

Proposition 3.6 and Proposition 3.8 together illustrate that VDPO can effectively improve the sample efficiency while guaranteeing consistent performance with SOTAs [20, 23, 42].

## 3.3 VDPO Implementation

In this section, we detail the implementations of VDPO, specifically the maximization Eq. (4) and the minimization Eq. (6) respectively. The pseudocode of VDPO is summarized in Alg. 1, and the training pipeline of VDPO is presented in Fig. 1.

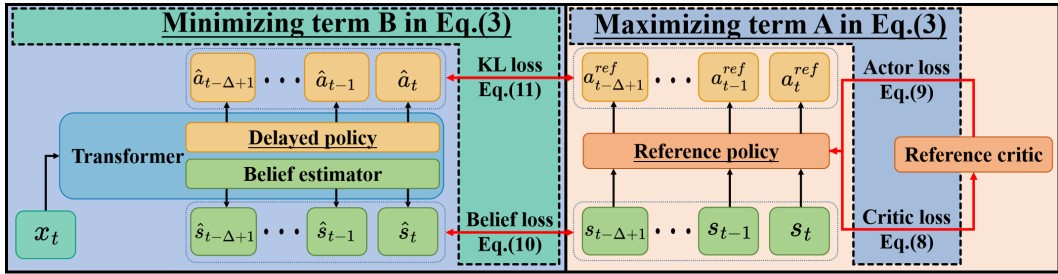

Figure 1: The training pipeline of VDPO.

---

**Algorithm 1** Variational Delayed Policy Optimization

---

**Input:** the reference policy $\pi_\psi$ and critic $Q_\theta$; transformer with belief head $b_\phi$ and policy head $\pi_\varphi$;
**for** each update step **do**
    # *A* of Eq. (3): Maximizing the performance of the reference policy $\pi$
    Updating critic $Q_\theta$ via Eq. (8) # Soft policy evaluation
    Updating policy $\pi_\psi$ via Eq. (9) # Soft policy improvement
    # *B* of Eq. (3): Minimizing the state-level KL between $\pi$ and $\pi_\triangle$
    Updating belief head $b_\phi$ via Eq. (10) # Belief representation learning
    Updating policy head $\pi_\varphi$ via Eq. (11) # Behaviour cloning
**end for**
**Output:** $b_\phi$ and $\pi_\varphi$

---

Eq. (4) aims to maximize the performance of the reference policy $\pi$ in the delay-free setting, which VDPO addresses using Soft Actor-Critic [13]. Specifically, given transition data $(s_t, a_t, r_t, s_{t+1})$, SAC updates the critic $Q_\theta$ parameterized by $\theta$ via minimizing the soft TD error:

$$\nabla_\theta \left[ \frac{1}{2} (Q_\theta(s_t, a_t) - \mathbb{Y})^2 \right], \tag{8}$$

where $\mathbb{Y} = r_t + \gamma \, \mathbb{E}_{a_{t+1} \sim \pi_\psi(\cdot|s_{t+1})} \left[ Q_\theta(s_{t+1}, a_{t+1}) - \log \pi_\psi(a_{t+1}|s_{t+1}) \right]$ where $\pi_\psi$ is the reference policy parameterized by $\psi$. And the reference policy $\pi_\psi$ is optimized by the gradient update:

$$\nabla_\psi \, \mathbb{E}_{\hat{a} \sim \pi_\psi(\cdot|s_t)} \left[ \log \pi_\psi(\hat{a}|s_t) - Q_\theta(s_t, \hat{a}) \right], \tag{9}$$

Eq. (6) aims to minimize the state-level KL divergence between the reference policy $\pi$ and delayed policy $\pi_\triangle$. Note that the true state $s_t$ under the delayed environment is inaccessible. Thus VDPO adopts a two-head transformer [37] to approximate not only the delayed policy $\pi_\triangle$, but also the belief estimator $b$ that predicts the state $\hat{s}_t$, as transformer shows a superior representation performance in behaviour cloning [8, 18]. We also discuss how different neural representations influence the RL performance later in Sec. 4.2.3. A similar transformer architecture proposed in [24] is adopted, which serializes the augmented state $x_t = \{s_{t-\delta}, a_{t-\delta}, \ldots, a_{t-1}\}$ to $\{(s_{t-\delta}, a_{t-i})\}_{i=\Delta}^1$ as the input. Based on the information bottleneck principle [34], the encoder needs to encode the input as embeddings with sufficient information related to the true states. Thus, the belief decoder and the policy decoder share a common encoder which is only trained while training the belief decoder, and we freeze the gradient backward of the encoder in training the policy decoder.

Specifically, for a given augmented state $x_t$ and true states $\{s_{t-\Delta+i}\}_{i=1}^\Delta$, the belief decoder $b_\phi$ parameterized by $\phi$ aims to reconstruct the states $\{s_{t-\Delta+i}\}_{i=1}^\Delta$ based on the $x_t$. Therefore, the belief decoder $b_\phi$ is optimized by the reconstruction loss:

$$\nabla_\phi \sum_{i=1}^\Delta \left[ \text{MSE}(b_\phi^{(i)}(x_t), s_{t-\Delta+i}) \right], \tag{10}$$

where $b_\phi^{(i)}(x_t)$ is the $i$-th reconstructed state of the belief decoder $b_\phi$ and MSE is the mean square error loss. Given the reference policy $\pi_\psi$ and the pair of augmented state and states $(x_t, \{s_{t-\Delta+i}\}_{i=1}^\Delta)$, the policy decoder $\pi_\varphi$ parameterized by $\varphi$ is optimized by minimizing the KL loss:

$$\nabla_\varphi \sum_{i=1}^\Delta \left[ \text{KL}(\pi_\varphi^{(i)}(\cdot|x_t) || \pi_\psi(\cdot|s_{t-\Delta+i})) \right], \tag{11}$$

where $\pi_\varphi^{(i)}(\cdot|x_t)$ is the $i$-th output of the policy decoder $\pi_\varphi$.

# 4 Experimental Results

## 4.1 Experiment Settings

We evaluate our VDPO in the MuJoCo benchmark [35]. For the selection of baselines, we choose the existing SOTAs including Augmented SAC (A-SAC) [13], DC/AC [5], DIDA [23], BPQL [20] and AD-SAC [42]. The setting of hyper-parameters is presented in Appendix A. We investigate the sample efficiency (Sec. 4.2.1) followed by performance comparison under different settings of delays (Sec. 4.2.2). We also conduct the ablation study on the representation of VDPO (Sec. 4.2.3). Each method was run over 10 random seeds. The training curves can be found in the Appendix E.

## 4.2 Experimental Results

### 4.2.1 Sample Efficiency

We first evaluate the sample efficiency in the MuJoCo with 5 constant delays. Using the performance of a delay-free policy trained by SAC, $Ret_{df}$, as the threshold, we report the required steps to reach this threshold within 1M global steps in Table 1. From the results, we can tell that VDPO shows strong superiority in terms of sample efficiency, successfully reaching the threshold in all tasks and achieving the best sample efficiency in 7 out of 9 tasks. Specifically, VDPO only requires 0.42M and 0.67M steps to reach the threshold in *Ant-v4* and *Humanoid-v4* respectively, while none of the baselines can reach the threshold within 1M steps. In *Halfcheetah-v4*, *Hopper-v4*, *Pusher-v4*, *Swimmer-v4* and *Walker2d-v4*, the steps taken by our VDPO is around 51% (ranging from 25% to 78%) of that required by AD-SAC, SOTA baseline. Based on these results, we can conclude that VDPO shows a significant advantage in sample complexity compared to other baselines. Additional experimental results of 25 and 50 constant delays are presented in Appendix D.

Table 1: Amount of steps required to reach the threshold $Ret_{df}$ in MuJoCo tasks with 5 constant delays within 1M global steps, where $\times$ denotes that failed to hit the threshold within 1M global steps. The best result is in blue.

| Task (Delays=5) | A-SAC | DC/AC | DIDA | BPQL | AD-SAC | VDPO (ours) |
|---|---|---|---|---|---|---|
| Ant-v4 | $\times$ | $\times$ | $\times$ | $\times$ | $\times$ | 0.42M |
| HalfCheetah-v4 | $\times$ | $\times$ | $\times$ | 0.99M | 0.56M | 0.44M |
| Hopper-v4 | 0.83M | 0.35M | $\times$ | 0.29M | 0.12M | 0.07M |
| Humanoid-v4 | $\times$ | $\times$ | $\times$ | $\times$ | $\times$ | 0.67M |
| HumanoidStandup-v4 | 0.64M | 0.35M | 0.10M | 0.09M | 0.14M | 0.14M |
| Pusher-v4 | 0.17M | 0.02M | 0.10M | 0.27M | 0.04M | 0.01M |
| Reacher-v4 | $\times$ | 0.61M | 0.10M | 0.90M | 0.44M | 0.77M |
| Swimmer-v4 | $\times$ | 0.94M | 0.10M | $\times$ | 0.13M | 0.07M |
| Walker2d-v4 | $\times$ | $\times$ | $\times$ | 0.52M | 0.67M | 0.25M |

### 4.2.2 Performance Comparison

The performance of VDPO and baselines are evaluated on MuJoCo with various settings and a normalized indicator [39, 42] $Ret_{nor} = \frac{Ret_{alg} - Ret_{rand}}{Ret_{df} - Ret_{rand}}$, where $Ret_{alg}$ and $Ret_{rand}$ are the performance of the algorithm and random policy, respectively. The results of MuJoCo benchmarks with 5, 25, and 50 constant delays are shown in the Table 2, showing that VDPO and AD-SAC outperform other baselines significantly in most tasks. Overall, VDPO and AD-SAC (SOTA) show a comparable performance, which is consistent with the theoretical observation in Sec. 3.2.

### 4.2.3 Additional Discussions

In this section, we conduct the ablation study to investigate the performance of VDPO using different neural representations. Furthermore, we explore whether VDPO is robust under stochastic delays.

**Ablation Study on Representations.**  As mentioned in Sec. 3.3, we investigate how the choice of neural representations for belief and policy influences the performance of VDPO. Baselines include multiple-layer perceptron (MLP) and Transformer without belief decoder. The results presented in Table 3 show that the two-head transformer used by our approach yields the best performance

Table 2: Normalized Performance $Ret_{nor}$ in MuJoCo tasks with 5, 25, and 50 constant delays for 1M global steps, where $\pm$ denotes the standard deviation. The best performance is in blue.

| Task | Delays | A-SAC | DC/AC | DIDA | BPQL | AD-SAC | VDPO (ours) |
|---|---|---|---|---|---|---|---|
| Ant-v4 | 5 | $0.18_{\pm0.01}$ | $0.25_{\pm0.05}$ | $0.89_{\pm0.03}$ | $0.96_{\pm0.03}$ | $0.72_{\pm0.25}$ | $1.11_{\pm0.04}$ |
| | 25 | $0.07_{\pm0.07}$ | $0.19_{\pm0.02}$ | $0.29_{\pm0.07}$ | $0.57_{\pm0.11}$ | $0.66_{\pm0.04}$ | $0.56_{\pm0.06}$ |
| | 50 | $0.02_{\pm0.04}$ | $0.19_{\pm0.02}$ | $0.19_{\pm0.05}$ | $0.38_{\pm0.07}$ | $0.48_{\pm0.06}$ | $0.46_{\pm0.07}$ |
| HalfCheetah-v4 | 5 | $0.35_{\pm0.15}$ | $0.40_{\pm0.23}$ | $0.90_{\pm0.01}$ | $1.00_{\pm0.06}$ | $1.07_{\pm0.06}$ | $1.03_{\pm0.08}$ |
| | 25 | $0.04_{\pm0.01}$ | $0.16_{\pm0.07}$ | $0.12_{\pm0.03}$ | $0.87_{\pm0.04}$ | $0.71_{\pm0.12}$ | $0.70_{\pm0.17}$ |
| | 50 | $0.12_{\pm0.17}$ | $0.12_{\pm0.13}$ | $0.15_{\pm0.03}$ | $0.73_{\pm0.17}$ | $0.74_{\pm0.10}$ | $0.72_{\pm0.21}$ |
| Hopper-v4 | 5 | $1.02_{\pm0.28}$ | $1.16_{\pm0.25}$ | $0.40_{\pm0.40}$ | $1.25_{\pm0.09}$ | $1.07_{\pm0.30}$ | $1.22_{\pm0.08}$ |
| | 25 | $0.13_{\pm0.04}$ | $0.19_{\pm0.04}$ | $0.27_{\pm0.08}$ | $1.21_{\pm0.18}$ | $0.86_{\pm0.25}$ | $0.82_{\pm0.40}$ |
| | 50 | $0.04_{\pm0.01}$ | $0.04_{\pm0.01}$ | $0.09_{\pm0.01}$ | $0.71_{\pm0.13}$ | $0.72_{\pm0.03}$ | $0.22_{\pm0.04}$ |
| Humanoid-v4 | 5 | $0.13_{\pm0.02}$ | $0.59_{\pm0.17}$ | $0.08_{\pm0.04}$ | $0.96_{\pm0.05}$ | $0.98_{\pm0.07}$ | $1.15_{\pm0.07}$ |
| | 25 | $0.05_{\pm0.01}$ | $0.04_{\pm0.01}$ | $0.07_{\pm0.00}$ | $0.12_{\pm0.01}$ | $0.25_{\pm0.16}$ | $0.12_{\pm0.02}$ |
| | 50 | $0.04_{\pm0.01}$ | $0.03_{\pm0.01}$ | $0.07_{\pm0.00}$ | $0.08_{\pm0.01}$ | $0.10_{\pm0.01}$ | $0.12_{\pm0.00}$ |
| HumanoidStandup-v4 | 5 | $1.02_{\pm0.08}$ | $1.16_{\pm0.12}$ | $1.00_{\pm0.00}$ | $1.13_{\pm0.07}$ | $1.22_{\pm0.03}$ | $1.29_{\pm0.02}$ |
| | 25 | $0.97_{\pm0.09}$ | $1.03_{\pm0.03}$ | $0.97_{\pm0.02}$ | $1.09_{\pm0.05}$ | $1.15_{\pm0.08}$ | $1.13_{\pm0.12}$ |
| | 50 | $0.90_{\pm0.02}$ | $1.02_{\pm0.07}$ | $0.89_{\pm0.06}$ | $1.06_{\pm0.04}$ | $1.12_{\pm0.02}$ | $1.04_{\pm0.16}$ |
| Pusher-v4 | 5 | $1.11_{\pm0.02}$ | $1.29_{\pm0.05}$ | $1.01_{\pm0.01}$ | $1.06_{\pm0.08}$ | $1.36_{\pm0.01}$ | $1.17_{\pm0.06}$ |
| | 25 | $0.49_{\pm0.32}$ | $1.12_{\pm0.02}$ | $1.04_{\pm0.01}$ | $1.07_{\pm0.06}$ | $1.29_{\pm0.03}$ | $1.31_{\pm0.07}$ |
| | 50 | $0.00_{\pm0.05}$ | $1.13_{\pm0.01}$ | $1.04_{\pm0.02}$ | $1.09_{\pm0.05}$ | $1.23_{\pm0.02}$ | $1.33_{\pm0.05}$ |
| Reacher-v4 | 5 | $0.97_{\pm0.01}$ | $1.02_{\pm0.00}$ | $1.03_{\pm0.00}$ | $1.00_{\pm0.01}$ | $1.03_{\pm0.01}$ | $1.02_{\pm0.03}$ |
| | 25 | $0.96_{\pm0.02}$ | $1.00_{\pm0.00}$ | $0.98_{\pm0.01}$ | $0.87_{\pm0.05}$ | $0.98_{\pm0.02}$ | $1.02_{\pm0.03}$ |
| | 50 | $0.86_{\pm0.02}$ | $0.89_{\pm0.01}$ | $0.93_{\pm0.02}$ | $0.90_{\pm0.02}$ | $0.91_{\pm0.03}$ | $1.02_{\pm0.03}$ |
| Swimmer-v4 | 5 | $0.88_{\pm0.09}$ | $1.11_{\pm0.30}$ | $1.05_{\pm0.01}$ | $0.97_{\pm0.02}$ | $1.82_{\pm0.78}$ | $2.30_{\pm0.36}$ |
| | 25 | $0.72_{\pm0.02}$ | $0.78_{\pm0.12}$ | $0.93_{\pm0.09}$ | $1.36_{\pm0.56}$ | $2.52_{\pm0.40}$ | $2.35_{\pm0.27}$ |
| | 50 | $0.69_{\pm0.04}$ | $0.68_{\pm0.06}$ | $0.87_{\pm0.03}$ | $2.23_{\pm0.55}$ | $2.71_{\pm0.14}$ | $2.42_{\pm0.22}$ |
| Walker2d-v4 | 5 | $0.76_{\pm0.21}$ | $0.85_{\pm0.12}$ | $0.61_{\pm0.07}$ | $1.20_{\pm0.11}$ | $1.12_{\pm0.09}$ | $1.27_{\pm0.04}$ |
| | 25 | $0.12_{\pm0.02}$ | $0.26_{\pm0.08}$ | $0.10_{\pm0.02}$ | $0.59_{\pm0.30}$ | $0.72_{\pm0.11}$ | $0.27_{\pm0.11}$ |
| | 50 | $0.11_{\pm0.02}$ | $0.11_{\pm0.02}$ | $0.08_{\pm0.01}$ | $0.23_{\pm0.10}$ | $0.23_{\pm0.11}$ | $0.11_{\pm0.03}$ |

compared to other candidates. The results also confirm that an explicit belief estimator implemented by a belief decoder can effectively improve performance.

Table 3: Normalized Performance $Ret_{nor}$ of VDPO using different representations in MuJoCo tasks with 5 constant delays, where $\pm$ denotes the standard deviation. The best result is in blue.

| Tasks (Delays=5) | MLP | Transformer w/o belief | Transformer w/ belief (ours) |
|---|---|---|---|
| Ant-v4 | $0.86_{\pm0.20}$ | $1.09_{\pm0.05}$ | $1.11_{\pm0.04}$ |
| HalfCheetah-v4 | $0.95_{\pm0.08}$ | $1.37_{\pm0.11}$ | $1.03_{\pm0.08}$ |
| Hopper-v4 | $1.11_{\pm0.19}$ | $1.13_{\pm0.29}$ | $1.22_{\pm0.08}$ |
| Humanoid-v4 | $0.78_{\pm0.11}$ | $0.89_{\pm0.43}$ | $1.15_{\pm0.07}$ |
| HumanoidStandup-v4 | $1.28_{\pm0.05}$ | $1.28_{\pm0.08}$ | $1.29_{\pm0.02}$ |
| Pusher-v4 | $1.35_{\pm0.04}$ | $1.34_{\pm0.05}$ | $1.17_{\pm0.06}$ |
| Reacher-v4 | $1.02_{\pm0.05}$ | $1.02_{\pm0.04}$ | $1.02_{\pm0.03}$ |
| Swimmer-v4 | $2.29_{\pm0.37}$ | $2.11_{\pm0.08}$ | $2.30_{\pm0.36}$ |
| Walker2d-v4 | $1.13_{\pm0.20}$ | $1.15_{\pm0.14}$ | $1.27_{\pm0.04}$ |

**Stochastic Delays.** We compare the performance in the MuJoCo with 5 stochastic delays where $\Delta = 5$ is with a probability of 0.9 and $\Delta \in [1, 5]$ is with a probability of 0.1. The results in the Table 4 demonstrate that VDPO outperforms other baselines at most tasks under stochastic delays. Especially in the *Ant-v4* and *Walker2d-v4*, VDPO performs approximately $62\%$ and $49\%$ better than the second best approach, respectively. In the *Reacher-v4* and *Swimmer-v4*, VDPO achieves a comparative performance with the best baseline. We will conduct a theoretical analysis of VDPO under stochastic delays in the future.

**Limitations and Future Works.** We mainly consider deterministic benchmarks in this paper, which are commonly adopted in the SOTAs [5, 23, 39]. However, the recent work ADRL [42] illustrates that existing approaches may have performance degeneration in stochastic environments, which can be mitigated by learning an auxiliary delayed task concomitantly. We will investigate in the future to integrate VDPO with ADRL to address stochastic applications.

Table 4: Normalized Performance $Ret_{nor}$ in MuJoCo tasks with 5 stochastic delays for 1M global steps, where $\pm$ denotes the standard deviation. The best result is in blue.

| Tasks | A-SAC | DC/AC | DIDA | BPQL | AD-SAC | VDPO (ours) |
|---|---|---|---|---|---|---|
| Ant-v4 | $0.18_{\pm0.01}$ | $0.27_{\pm0.02}$ | $0.55_{\pm0.08}$ | $0.58_{\pm0.12}$ | $0.69_{\pm0.17}$ | $1.12_{\pm0.04}$ |
| HalfCheetah-v4 | $0.36_{\pm0.12}$ | $0.36_{\pm0.18}$ | $0.75_{\pm0.02}$ | $0.76_{\pm0.16}$ | $1.03_{\pm0.06}$ | $1.07_{\pm0.09}$ |
| Hopper-v4 | $0.85_{\pm0.22}$ | $0.94_{\pm0.29}$ | $0.31_{\pm0.08}$ | $0.68_{\pm0.34}$ | $1.05_{\pm0.22}$ | $1.35_{\pm0.11}$ |
| Humanoid-v4 | $0.15_{\pm0.06}$ | $0.67_{\pm0.18}$ | $0.07_{\pm0.01}$ | $0.40_{\pm0.42}$ | $0.97_{\pm0.07}$ | $1.06_{\pm0.00}$ |
| HumanoidStandup-v4 | $1.03_{\pm0.05}$ | $1.20_{\pm0.08}$ | $1.00_{\pm0.00}$ | $1.10_{\pm0.07}$ | $1.26_{\pm0.07}$ | $1.27_{\pm0.01}$ |
| Pusher-v4 | $1.11_{\pm0.02}$ | $1.17_{\pm0.02}$ | $1.02_{\pm0.01}$ | $1.07_{\pm0.05}$ | $1.22_{\pm0.01}$ | $1.34_{\pm0.05}$ |
| Reacher-v4 | $0.98_{\pm0.01}$ | $1.02_{\pm0.01}$ | $1.02_{\pm0.00}$ | $0.85_{\pm0.11}$ | $1.05_{\pm0.01}$ | $1.01_{\pm0.04}$ |
| Swimmer-v4 | $0.82_{\pm0.10}$ | $1.47_{\pm0.58}$ | $1.03_{\pm0.02}$ | $1.53_{\pm0.52}$ | $2.36_{\pm0.64}$ | $2.13_{\pm0.18}$ |
| Walker2d-v4 | $0.68_{\pm0.28}$ | $0.89_{\pm0.08}$ | $0.54_{\pm0.09}$ | $0.59_{\pm0.30}$ | $0.63_{\pm0.39}$ | $1.33_{\pm0.11}$ |

## 5 Related Works

Compared to the common delay-free setting, delayed RL with disrupted Markovian property [4, 19] is closer to real-world complex applications, such as robotics [17, 26], transportation systems [6] and financial market trading [14]. Existing delayed RL techniques conduct learning over either original state space (referred to as direct approach) or augmented state space (referred to as augmentation-based approach). Direct approaches enjoy high learning efficiency by learning in the original small state space. However, early approaches simply ignore the absence of Markovian property caused by delay and directly conduct classical RL techniques based on delayed observations, which distinctly suffer from serious performance drops. The subsequential improvement is to train based on unobserved instant observations, which are predicted by various generative models, e.g., deterministic generative models [38], Gaussian distributions [7], and transformers [24]. However, the inherent approximation errors in these learned models introduce prediction inaccuracy and result in sub-optimal performance issues [24]. To summarize, direct approaches achieve high learning efficiency, but commonly with a cost of performance degeneration.

The augmentation-based approach is notably more promising as it retrieves Markovian property via augmenting the state with the actions related to delays and thus legitimately enables RL techniques over the yielded delayed MDP [4, 19]. However, the augmentation-based approach works in a significantly larger state space, which is thus plagued by the curse of dimensionality, resulting in learning inefficiency. To mitigate this issue, DC/AC [5] leverages the multi-step off-policy technique to develop a partial trajectory resampling operator to accelerate the learning process. Based on the dataset aggregation technique, DIDA [23] generalizes the pre-trained delay-free policy into an augmented policy. Recent attempts [20, 39] evaluate the augmented policy by a non-augmented Q-function for improving learning efficiency. ADRL [42] suggests introducing an auxiliary delayed task with changeable auxiliary delays for the trade-off between the learning efficiency and performance degeneration in the stochastic MDP. However, these approaches still suffer from the sample complexity issue due to the fundamental challenge of TD learning in high dimensional state space.

The conceptualization of RL as an inference problem has gained attraction recently, allowing the adaption of various optimization tricks to enhance RL efficiency [11, 15, 21, 31]. For instance, VIP [30] integrates different projection techniques into the policy search approach based on variational inference. Virel [9] introduces a variational inference framework that reduces the actor-critic method to the Expectation-Maximization (EM) algorithm. MPO [1, 2] is a family of off-policy entropy-regularized methods in the EM fashion. CVPO [25] extends MPO to the safety-critical settings. The novel trial in this work of viewing the delayed RL as a variational inference problem allows us to use extensive optimization tools to address the sample complexity issue in delayed RL.

## 6 Conclusion

This work explores the challenges of RL problems in environments with inherent delays between agent interactions and their consequences. Existing delayed RL methods often suffer from learning inefficiency as temporal-difference learning in the delayed MDP with high dimensional augmented state space demands an increased sample size. To address this limitation, we present VDPO, a new delayed RL approach rooted in the variational inference principle. VDPO redefines the delayed RL problem into a two-step iterative optimization problem. It alternates between (1) maximizing the

performance of the reference policy by temporal-difference learning in the delay-free setting and (2) minimizing the KL divergence between the reference and delayed policies by behaviour cloning. Furthermore, our theoretical analysis and the empirical results in the MuJoCo benchmark validate that VDPO not only effectively improves the sample efficiency but also maintains a robust performance level.

## Acknowledgments and Disclosure of Funding

We sincerely acknowledge the support by the grant EP/Y002644/1 under the EPSRC ECR International Collaboration Grants program, funded by the International Science Partnerships Fund (ISPF) and the UK Research and Innovation, and the grant 2324936 by the US National Science Foundation. This work is also supported by Taiwan NSTC under Grant Number NSTC-112-2221-E-002-168-MY3.

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

# A  Implementation Details

The hyper-parameters setting of VDPO is presented in Table 5. For baselines, we adopt similar hyper-parameters settings as suggested by original works, including A-SAC, DC/AC [5], DIDA [23], BPQL [20] and AD-SAC [42]. The implementation of VDPO is based on CleanRL [16], and we also provide the code and guidelines to reproduce our results in the supplemental material. Each run of VDPO takes approximately 6 hours on 1 NVIDIA A100 GPU and 8 Intel Xeon CPUs.

Table 5: Hyper-parameters table of VDPO.

| Hyper-parameter | Value |
|---|---|
| Buffer Size | 1,000,000 |
| Batch Size | 256 |
| Global Timesteps | 1,000,000 |
| Discount Factor | 0.99 |
| Reference Policy | |
| Learning Rate for Actor | 3e-4 |
| Learning Rate for Critic | 1e-3 |
| Network Layers | 3 |
| Network Neurons | [256, 256] |
| Activation | ReLU |
| Optimizer | Adam |
| Initial Entropy | $-|\mathcal{A}|$ |
| Learning Rate for Entropy | 1e-3 |
| Train Frequency for Actor | 2 |
| Train Frequency for Critic | 1 |
| Soft Update Factor for Critic | 5e-3 |
| Delayed Policy (Transformer) | |
| Sequence Length | $\Delta$ |
| Embedding Dimension | 256 |
| Attention Heads | 1 |
| Layers Num | 3 |
| Attention Dropout Rate | 0.1 |
| Residual Dropout Rate | 0.1 |
| Embedding Dropout Rate | 0.1 |
| Learning Rate | 3e-4 |
| Optimizer | Adam |
| Train Frequency for Belief Decoder | 1 |
| Train Frequency for Policy Decoder | 1,000 |

# B   Evidence Lower BOund (ELBO) of Delayed RL

Following the similar derivation sketch with [2, 25], we provide the derivation of Eq. (2) as follows.

$$
\begin{aligned}
\log p_{\pi_\Delta}(O = 1) &= \log \int p(O = 1|\tau)p_{\pi_\Delta}(\tau)\mathrm{d}\tau \\
&= \log \int p_\pi(\tau)\frac{p(O = 1|\tau)p_{\pi_\Delta}(\tau)}{p_\pi(\tau)}\mathrm{d}\tau \\
&\geq \int p_\pi(\tau)\log\left[\frac{p(O = 1|\tau)p_{\pi_\Delta}(\tau)}{p_\pi(\tau)}\right]\mathrm{d}\tau \\
&= \mathbb{E}_{\tau\sim p_\pi(\tau)}\left[\log p(O = 1|\tau)\right] - \mathrm{KL}(p_\pi(\tau)||p_{\pi_\Delta}(\tau)).
\end{aligned}
$$

# C   Theoretical Analysis

**Proposition C.1** (State-level KL divergence). *For a fixed reference policy $\pi$, the trajectory-level KL divergence can transform into the formulation of state-level KL divergence as follows*

$$
\mathrm{KL}(p_\pi(\tau)||p_{\pi_\Delta}(\tau)) = \underbrace{\sum_{t=0}^{\infty}\int d^\pi(s_t)\mathrm{KL}(\pi(a_t|s_t)||\pi_\Delta(a_t|x_t))\mathrm{d}s_t}_{\text{State-level KL divergence}} + Const.. \tag{12}
$$

*where* $Const. = \mathrm{KL}(\rho(s_0)||\rho_\Delta(x_0))$
$$
+ \sum_{t=0}^{\infty}\int d^\pi(s_t)\int\pi(a_t|s_t)\mathrm{KL}(\mathcal{P}(s_{t+1}|s_t, a_t)||b(s_t|x_t)\mathcal{P}_\Delta(x_{t+1}|x_t, a_t))\mathrm{d}a_t\mathrm{d}s_t.
$$

*Proof.* The trajectory distribution $p_\pi(\tau)$ induced by $\pi$ is given by:

$$
p_\pi(\tau) = \rho(s_0)\prod_{t=0}^{\infty}P(s_{t+1}|s_t, a_t)\pi(a_t|s_t).
$$

Similarly, the trajectory distribution $p_{\pi_\Delta}(\tau)$ induced by $\pi_\Delta$ is given by:

$$
p_{\pi_\Delta}(\tau) = \rho_\Delta(x_0)b(s_0|x_0)\prod_{t=0}^{\infty}b(s_{t+1}|x_{t+1})\mathcal{P}_\Delta(x_{t+1}|x_t, a_t)\pi_\Delta(a_t|x_t).
$$

Therefore, the trajectory-level KL divergence can be written as

$$
\begin{aligned}
&\mathrm{KL}(p_\pi(\tau)||p_{\pi_\Delta}(\tau)) \\
&= \mathbb{E}_{\tau\sim p_\pi(\tau)}\left[\log p_\pi(\tau) - \log p_{\pi_\Delta}(\tau)\right] \\
&= \mathbb{E}_{\tau\sim p_\pi(\tau)}\Big[\log p(s_0) + \sum_{t=0}^{\infty}\left[\log\left[\mathcal{P}(s_{t+1}|s_t, a_t)\pi(a_t|s_t)\right]\right] \\
&\quad - \log\left[p(x_0)b(s_0|x_0)\right] - \sum_{t=0}^{\infty}\left[\log b(s_{t+1}|x_{t+1}) + \log P_\Delta(x_{t+1}|x_t, a_t) + \log\pi_\Delta(a_t|x_t)\right]\Big] \\
&= \mathbb{E}_{\tau\sim p_\pi(\tau)}\left[\sum_{t=0}^{\infty}\left[\log\mathcal{P}(s_{t+1}|s_t, a_t) + \log\pi(a_t|s_t)\right] - \sum_{t=0}^{\infty}\left[\log b(s_t|x_t) + \log\mathcal{P}_\Delta(x_{t+1}|x_t, a_t) + \log\pi_\Delta(a_t|x_t)\right]\right] \\
&= \mathbb{E}_{\tau\sim p_\pi(\tau)}\left[\log\left[\frac{\rho(s_0)}{\rho_\Delta(x_0)}\right] + \sum_{t=0}^{\infty}\left[\log\frac{\mathcal{P}(s_{t+1}|s_t, a_t)}{b(s_t|x_t)\mathcal{P}_\Delta(x_{t+1}|x_t, a_t)} + \log\frac{\pi(a_t|s_t)}{\pi_\Delta(a_t|x_t)}\right]\right] \\
&= \underbrace{\mathbb{E}_{\tau\sim p_\pi(\tau)}\left[\log\left[\frac{\rho(s_0)}{\rho_\Delta(x_0)}\right] + \sum_{t=0}^{\infty}\left[\log\frac{\mathcal{P}(s_{t+1}|s_t, a_t)}{b(s_t|x_t)\mathcal{P}_\Delta(x_{t+1}|x_t, a_t)}\right]\right]}_{C} + \underbrace{\mathbb{E}_{\tau\sim p_\pi(\tau)}\left[\sum_{t=0}^{\infty}\left[\log\frac{\pi(a_t|s_t)}{\pi_\Delta(a_t|x_t)}\right]\right]}_{D}.
\end{aligned}
$$

For $C$, we have

$$C = \text{KL}(\rho(s_0)||\rho_\Delta(x_0)) + \sum_{t=0}^\infty \int p_\pi(\tau) \log \frac{\mathcal{P}(s_{t+1}|s_t, a_t)}{b(s_t|x_t)\mathcal{P}_\Delta(x_{t+1}|x_t, a_t)} d\tau$$

$$= \text{KL}(\rho(s_0)||\rho_\Delta(x_0)) + \sum_{t=0}^\infty \int d^\pi(s_t) \int \pi(a_t|s_t) \int \mathcal{P}(s_{t+1}|s_t, a_t) \log \frac{\mathcal{P}(s_{t+1}|s_t, a_t)}{b(s_t|x_t)\mathcal{P}_\Delta(x_{t+1}|x_t, a_t)} ds_{t+1} da_t ds_t$$

$$= \text{KL}(\rho(s_0)||\rho_\Delta(x_0)) + \sum_{t=0}^\infty \int d^\pi(s_t) \int \pi(a_t|s_t) \text{KL}(\mathcal{P}(s_{t+1}|s_t, a_t)||b(s_t|x_t)\mathcal{P}_\Delta(x_{t+1}|x_t, a_t)) da_t ds_t$$

$$\geq 0.$$

Therefore, $C$ is a constant determined by the transition function of the dynamics and the fixed reference policy $\pi$.

Then, for $D$, we have

$$D = \sum_{t=0}^\infty \int p_\pi(\tau) \log \frac{\pi(a_t|s_t)}{\pi_\Delta(a_t|x_t)} d\tau$$

$$= \sum_{t=0}^\infty \int d^\pi(s_t) \int \pi(a_t|s_t) \log \frac{\pi(a_t|s_t)}{\pi_\Delta(a_t|x_t)} da_t ds_t$$

$$= \sum_{t=0}^\infty \int d^\pi(s_t) \text{KL}(\pi(a_t|s_t)||\pi_\Delta(a_t|x_t)) ds_t.$$

$\square$

**Lemma C.2** (Sample complexity of VDPO). *Assumed that maximizing $A$ in Eq. (3) by model-based policy iteration while minimizing $B$ in Eq. (3) by behaviour cloning, VDPO finds an $\epsilon$-optimal policy with probability $1 - \sigma$ using sample size*

$$\mathcal{O}\left(\max\left(\frac{|\mathcal{S}||\mathcal{A}|}{(1-\gamma)^3\epsilon^2}\ln\frac{1}{\sigma}, \frac{|\mathcal{X}|\ln|\mathcal{A}|}{(1-\gamma)^4\epsilon^2}\sigma\right)\right).$$

*Proof.* Applying Lem. 3.2 and Lem. 3.4. $\square$

**Proposition C.3** (Sample complexity comparison). *In the delayed MDP, as $\sigma \to 0$, the sample complexity of VDPO (Lem. 3.5) is less or equal to the sample complexity of model-based policy iteration (Lem. 3.2):*

$$\mathcal{O}\left(\max\left(\frac{|\mathcal{S}||\mathcal{A}|}{(1-\gamma)^3\epsilon^2}\ln\frac{1}{\sigma}, \frac{|\mathcal{X}|\ln|\mathcal{A}|}{(1-\gamma)^4\epsilon^2}\sigma\right)\right) \leq \mathcal{O}\left(\frac{|\mathcal{X}||\mathcal{A}|}{(1-\gamma)^3\epsilon^2}\ln\frac{1}{\sigma}\right).$$

*Proof.* It is obvious that

$$\frac{|\mathcal{S}||\mathcal{A}|}{(1-\gamma)^3\epsilon^2}\ln\frac{1}{\sigma} \leq \frac{|\mathcal{X}||\mathcal{A}|}{(1-\gamma)^3\epsilon^2}\ln\frac{1}{\sigma},$$

as we have $|\mathcal{S}| \leq |\mathcal{X}| = |\mathcal{S}||\mathcal{A}|^\Delta$.

Then, we show that

$$\frac{|\mathcal{X}|\ln|\mathcal{A}|}{(1-\gamma)^4\epsilon^2}\sigma \leq \frac{|\mathcal{X}||\mathcal{A}|}{(1-\gamma)^3\epsilon^2}\ln\frac{1}{\sigma},$$

which is equivalent to

$$\frac{\ln|\mathcal{A}|}{|\mathcal{A}|} \leq -(1-\gamma)\frac{\ln\sigma}{\sigma}.$$

This inequality always holds when $\sigma \to 0$ as

$$\lim_{\sigma\to 0} -(1-\gamma)\frac{\ln\sigma}{\sigma} = +\infty \gg \frac{1}{e} > \frac{\ln|\mathcal{A}|}{|\mathcal{A}|}.$$

$\square$

**Lemma C.4** (Convergence of delayed policy in VDPO). *Let $\pi^*$ be the optimal reference policy which is trained by a delay-free RL algorithm. The delayed policy $\pi_\Delta$ converges to $\pi_\Delta^*$ satisfying that*

$$\pi_\Delta^*(a_t|x_t) = \mathbb{E}_{s_t \sim b(\cdot|x_t)}\left[\pi^*(a_t|s_t)\right], \forall x_t \in \mathcal{X}. \tag{13}$$

*Proof.* We can derive the result from the solution of Eq. (6). $\qquad\square$

**Proposition C.5** (Consistent fixed point). *VDPO shares the same fixed point (Eq. (7)) with DIDA [23], BPQL [20] and AD-SAC [42] for the same delayed MDP.*

*Proof.* The fixed points of DIDA (Eq. (3) in [23]), BPQL (Eq. (23) in [20]) and AD-SAC (Theorem 5.9 in [42]) all are

$$\pi_\Delta^*(a_t|x_t) = \mathbb{E}_{s_t \sim b(\cdot|x_t)}\left[\pi^*(a_t|s_t)\right], \forall x_t \in \mathcal{X}.$$

which is consistent with the fixed point of VDPO. $\qquad\square$

# D  Additional Experimental Results

In MuJoCo with 25 and 50 constant delays, We report the required steps to hit this threshold within 1M global steps in Table 6 and Table 7 respectively.

Table 6: Amount of steps required to hit the threshold $Ret_{df}$ in MuJoCo tasks with 25 constant delays within 1M global steps, where $\times$ denotes that failed to hit the threshold within 1M global steps. The best result is in blue.

| Task (Delays=25) | A-SAC | DC/AC | DIDA | BPQL | AD-SAC | VDPO (ours) |
|---|---|---|---|---|---|---|
| Ant-v4 | $\times$ | $\times$ | $\times$ | $\times$ | $\times$ | $\times$ |
| HalfCheetah-v4 | $\times$ | $\times$ | $\times$ | $\times$ | $\times$ | $\times$ |
| Hopper-v4 | $\times$ | $\times$ | $\times$ | 0.69M | $\times$ | $\times$ |
| Humanoid-v4 | $\times$ | $\times$ | $\times$ | $\times$ | $\times$ | $\times$ |
| HumanoidStandup-v4 | $\times$ | 0.38M | $\times$ | 0.09M | 0.09M | 0.48M |
| Pusher-v4 | $\times$ | 0.09M | 0.10M | 0.02M | 0.03M | 0.01M |
| Reacher-v4 | $\times$ | 0.83M | $\times$ | $\times$ | $\times$ | 0.22M |
| Swimmer-v4 | $\times$ | $\times$ | $\times$ | 0.39M | 0.12M | 0.09M |
| Walker2d-v4 | $\times$ | $\times$ | $\times$ | $\times$ | $\times$ | $\times$ |

Table 7: Amount of steps required to hit the threshold $Ret_{df}$ in MuJoCo tasks with 50 constant delays within 1M global steps, where $\times$ denotes that failed to hit the threshold within 1M global steps. The best result is in blue.

| Task (Delays=50) | A-SAC | DC/AC | DIDA | BPQL | AD-SAC | VDPO (ours) |
|---|---|---|---|---|---|---|
| Ant-v4 | $\times$ | $\times$ | $\times$ | $\times$ | $\times$ | $\times$ |
| HalfCheetah-v4 | $\times$ | $\times$ | $\times$ | $\times$ | $\times$ | $\times$ |
| Hopper-v4 | $\times$ | $\times$ | $\times$ | $\times$ | $\times$ | $\times$ |
| Humanoid-v4 | $\times$ | $\times$ | $\times$ | $\times$ | $\times$ | $\times$ |
| HumanoidStandup-v4 | $\times$ | 0.68M | $\times$ | 0.21M | 0.08M | 0.84M |
| Pusher-v4 | $\times$ | 0.14M | 0.10M | 0.18M | 0.02M | 0.01M |
| Reacher-v4 | $\times$ | $\times$ | $\times$ | $\times$ | $\times$ | 0.39M |
| Swimmer-v4 | $\times$ | $\times$ | $\times$ | 0.29M | 0.11M | 0.15M |
| Walker2d-v4 | $\times$ | $\times$ | $\times$ | $\times$ | $\times$ | $\times$ |

# E  Learning Curves

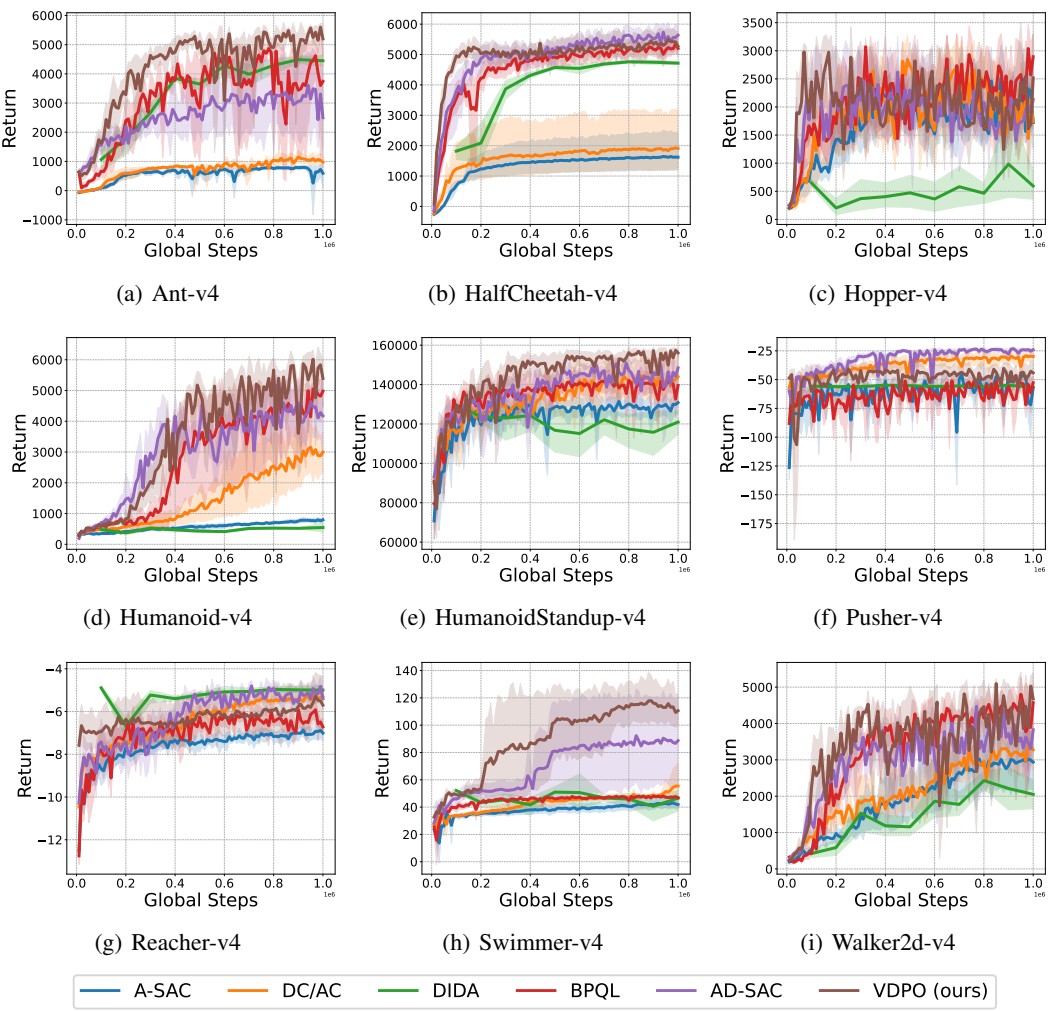

Figure 2: Learning curves in MuJoCo tasks with 5 constant delays where the shaded areas represented the standard deviation.

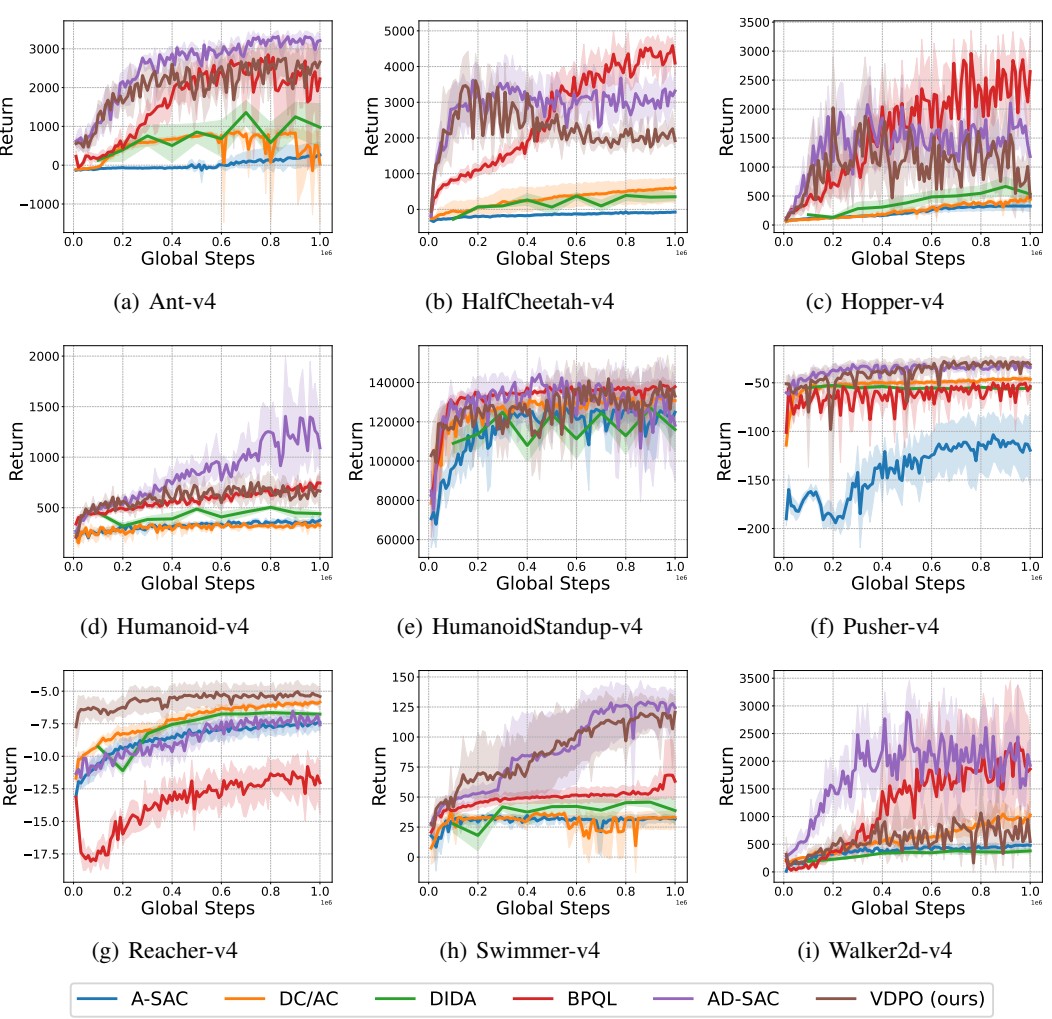

Figure 3: Learning curves in MuJoCo tasks with 25 constant delays where the shaded areas represented the standard deviation.

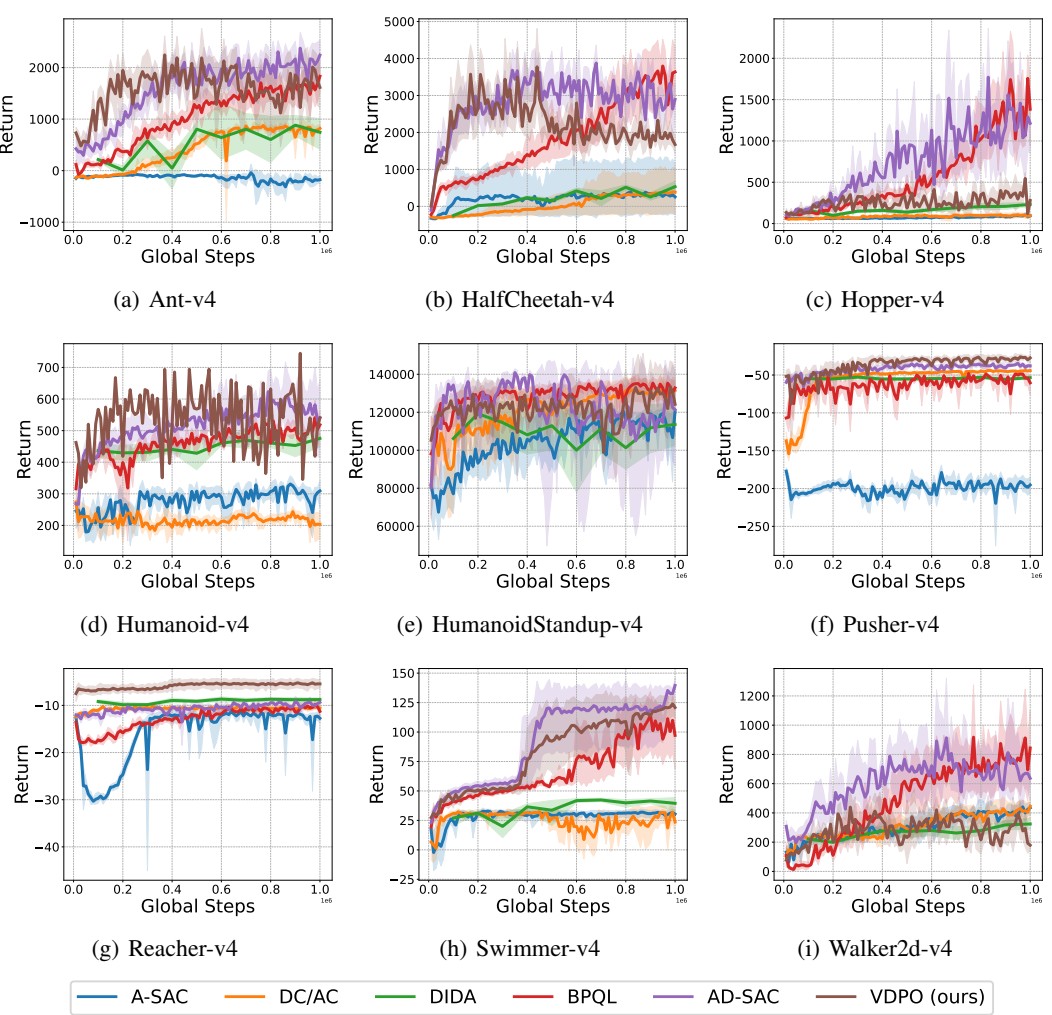

Figure 4: Learning curves in MuJoCo tasks with 50 constant delays where the shaded areas represented the standard deviation.

