# OpenReview forum: "Variational Delayed Policy Optimization"
_NeurIPS.cc/2024/Conference — NeurIPS 2024 spotlight_

### Official Review · Reviewer_io6Z · 2024-07-10

**Soundness:** 4
**Presentation:** 4
**Contribution:** 3
**Rating:** 8
**Confidence:** 4

**Summary:**

The paper re-examines the delayed RL framework in terms of an equivalent variational inference problem. The authors theoretically motivate maximising the return of a reference policy that does not feature delays using an argument of improved sample complexity and performance that improves with smaller delays. They then derive  an objective for behavioural cloning to learn this desired policy. These two optimisation problems amount to a variational EM style algorithm on the ELBO.

The authors then provide a theoretical analysis of their approach, deriving SOTA sample complexity and performance bounds. The authors then introduce their VDPO algorithm to practically carry out the VEM optimisation before evaluating in several domains where it achieves SOTA sample complexity and similar performance to the SOTA algorithm AD-SAC.

**Strengths:**

**Presentation and Clarity**

The writing style is clear, scientific and presentation is excellent. The authors do a great job summarising related work, the theoretical framework (I particularly liked how Eq 3 is presented and referred back to) and explaining their theoretical results carefully. The material is not simple and could become unwieldily in the wrong hands, so its a credit to the authors that they can present their work this way.

**Contribution**

The paper combines both insightful theoretical results with a solid empirical evaluation, making valid contributions to both. Their algorithm should be a reference benchmark for delayed RL and I can imagine their research having high impact in the field.

**Soundness**

I've checked the maths and everything follows clearly to me.

**Weaknesses:**

There are no major weaknesses of the paper.

There are a few typos that I found:

Line 74: $p\rightarrow \rho$

**Questions:**

I have no questions about the work

**Limitations:**

There are no major limitations and all negative effects are addressed.

---

> ### Author Rebuttal · Authors · 2024-08-05
>
> We genuinely appreciate the positive feedback provided by Reviewer io6Z.
> The typos will be corrected in the revised version.
> This work aims to present a new delayed RL method from the perspective of variational inference which can effectively improve the sample complexity without compromising the performance, supported by comprehensive theoretical analysis.
> Specifically, Equation 3 indeed plays a critical role in our framework. It bridges delayed RL and variational RL, which provides us the opportunity to use extensive optimization tools to solve the delayed RL problem effectively.
> Furthermore, Equation 3 inspires our delayed RL method, VDPO and its practical implementation.
> The experimental results demonstrate that our VDPO can effectively improve the sample efficiency while achieving comparable performance with the SOTA baseline.

---

> > ### Comment · Reviewer_io6Z · 2024-08-11
> > **Update**
> >
> > I thank the reviewers for their rebuttal and maintain my score. As other reviewers also seem to be in agreement, I strongly advocate for this paper's acceptance.

---

> > > ### Author Response · Authors · 2024-08-12
> > >
> > > We sincerely appreciate you for reviewing and giving insightful comments on our manuscript.

---

### Official Review · Reviewer_wuNr · 2024-07-11

**Soundness:** 3
**Presentation:** 4
**Contribution:** 3
**Rating:** 8
**Confidence:** 3

**Summary:**

The paper proposes a new algorithm (VDPO) for delayed RL which first learns a reference policy in the delay-free MDP using standard RL tools and then uses behavior cloning to encourage visitation of optimal trajectories in the delayed MDP.
The paper shows theoretical guarantees on sample complexity and performance, and conducts an extensive empirical evaluation where VDPO improves sample-efficiency upon SOTA.

**Strengths:**

* The proposed algorithm is well-motivated under the inference interpretation of RL and can be expressed succinctly in a natural objective.
* I find the approach of VDPO to address the sample complexity issue of baselines by first optimizing the reward in the delay-free MDP and then using behavior cloning very neat.
* The paper derives guarantees on sample complexity and performance of VDPO which compare favorably to those of naïve approaches using augmented state spaces.
* The experimental evaluation is extensive and shows a clear improvement on sample complexity.
* I find that the manuscript reads very well and is nicely structured.

**Weaknesses:**

* It would be interesting to see an extension of this approach to stochastic delays.

**Questions:**

None.

**Limitations:**

See weaknesses above.

---

> ### Author Rebuttal · Authors · 2024-08-05
>
> We sincerely appreciate the positive comments provided by Reviewer wuNr, and our responses are as follows.
>
> # Weakness 1: It would be interesting to see an extension of this approach to stochastic delays.
>
> We appreciate that the importance of addressing stochastic delays is recognized. Though this paper mainly focuses on constant delays, we also conduct experiments to explore the robustness of VDPO under stochastic delays, as shown in Table 4. However, the neat theoretical results under constant delays no longer hold when the delays are stochastic.
> For instance, when delays become stochastic, Lemma 3.1 does not hold as it is difficult to compare the performance between two stochastic delayed MDPs with different distributions of delays.
> Intuitively, learning in the delayed MDP with stochastic delays is challenging because the agent needs to learn the policy in the augmented state space with varying delays.
> Therefore, considerable effort is needed to derive theoretical results in this setting.
>
> We believe a significant modification is needed to adapt VDPO to stochastic delays and we will address it in our future work.

---

> > ### Comment · Reviewer_wuNr · 2024-08-08
> >
> > I thank the authors for their elaboration. After reading the other reviews and rebuttals, I am sticking to my current score.

---

> > > ### Author Response · Authors · 2024-08-09
> > >
> > > Sincerely appreciate your time and effort in reviewing and giving insightful comments on our manuscript.

---

### Official Review · Reviewer_FFJG · 2024-07-12

**Soundness:** 3
**Presentation:** 3
**Contribution:** 3
**Rating:** 6
**Confidence:** 4

**Summary:**

The paper proposes a novel delayed rl algorithm called variational delayed policy optimization, which reformulates delayed RL as a variational inference problem and solves it with a two-step iterative optimization. Both theoretical and empirical results show that VDPO achieves better performance in sample efficiency in the mujoco benchmark.

**Strengths:**

1.	The proposed method bridges the gap between delayed RL and variational RL, which shows its novelty.
2.	This paper presents both theoretical analysis and empirical results on the sample complexity of VDPO.
3.	The paper is overall well-written, and the idea is straightforward.

**Weaknesses:**

1.	It seems that the proposed method must be trained in a delayed-free environment. It may result in limited application scenarios.
2.	The proposed method utilizes transformer as the policy. Given the computational cost of transformer, it may result in longer delay in real-world applications. However, the authors do not take it into account in the experiments.
3.	Equation 10 seems based on the underlying assumption that the subsequence state obeys Gaussian distribution, which may not hold in some real-world scenarios.
4.	According to Figure 3, when the number of the constant delays is large, it looks that VDPO doesn’t work very well. Moreover, given the computational cost of transformer, the real performance of VDPO is doubtful.

**Questions:**

1.	It seems that the proposed method must be trained in a delayed-free environment. So the reviewer wonders whether the baseline methods like A-SAC are conducted in such setting. If they are not, the comparison is not quite fair from the reviewer’s perspective.
2.	Could the author present the comparison of inference time among VDPO and other baseline methods?

**Limitations:**

The authors have clearly presented the limitations in the paper.

---

> ### Author Rebuttal · Authors · 2024-08-05
>
> We thank Reviewer FFJG for the comment. Before replying to all the concerns and questions in detail, we want to clarify that this work adopts commonly used evaluation metrics, sample efficiency and performance (return), and conducts fair comparison with existing works [1, 2, 3]. Our detailed responses are as follows:
> # Weakness 1 and Question 1: Training in the delay-free environment and baseline selection.
> We appreciate the reviewer for the interesting questions. For **Weakness 1**, we clarify that **training in a delay-free environment performs no restriction on VDPO's applicability** and the reason is as below.
>
> Let $\{(s_{t-\Delta},a_t,r_t)\}$ be a sampled trajectory in an environment with a constant delay $\Delta$. At time step $t$, though the agent can only observe $s_{t-\Delta}$, the true instant state $s_{t}$ will be observed at $t+\Delta$. Thus we can easily synthesize a delay-free trajectory for training based on a time-delay trajectory. It illustrates why VDPO, the same as SOTAs [1, 2, 3] that also leverage the training in a delay-free environment, can be applied in any constant-delay applications.
>
> For **Question 1**, the SOTA DIDA [1], BPQL [2] and AD-SAC [3] all include delay-free training in their frameworks, which are the baseline methods in this paper. Sample efficiency and performance are the common evaluation metrics used in delay RL materials [1, 2, 3]. Therefore, we believe we conduct **fair comparison under common setting**, following the standard manner in delayed RL research.
>
> # Weakness 2 and Question 2: The inference delays of transformer in VDPO.
> We clarify that existing delayed RL works [1, 2, 3], including this paper, all use **sample efficiency and performance** as the evaluation metrics. The reason for not considering inference time is as below.
>
> The delays caused by the inference of the neural networks with common architectures are usually much less than the time delays in real-world applications (see Table R1). Specifically, in this work, the inference time of baselines (MLP) is around 0.5 ms, and the inference time of VDPO (transformer) is approximately 1.8 ms (all run on one NVIDIA A100 GPU), while in the teleoperation of robotics [4], especially in space robotics [5, 6], the delays are ranging from 5 s to 10 s. The inference time of the neural network is approximately **three orders of magnitude less than the delays**. Therefore, the inference delays can be safely neglected.
>
> Table R1: Inference time of different neural network architectures.
> |Delays=5|MLP|Transformer|
> |-|-|-|
> |Inference time (ms)|$0.529 (\pm 0.005)$|$1.858 (\pm 0.039)$|
>
> Additionally, if a significantly complex neural architecture is adopted in the future, the inference delay issue can be addressed naively by adding extra delay step in the training process in advance. For example, the policy can be trained in the delayed environment with 6 delays for deploying in the real environment with 5 delays, if the inference delay takes more or less 1 control cycle, since the inference time should be constant for a specific neural architecture under the same hardware setting.
>
> To sum up, we believe the inference delay issue is not a critical issue and can be easily addressed if necessary. However, we appreciate the reviewer for raising it and will add the discussion in the revised version to avoid any potential confusion.
>
> # Weakness 3: Assumption of Equation 10.
> We clarify that Equation 10 **does not** rely on the assumption that the subsequence state obeys Gaussian distribution. Equation 10 is the loss function of the belief function $b$ which aims to learn a representation for the delayed policy $\pi_\Delta$ via reconstructing the states $\\{s_{t-\Delta+i}\\}_{i=1}^\Delta$ from the augmented state $x_t$, inspired by existing works [7, 8].
>
> # Weakness 4: Performance of VDPO.
> In this work, VDPO aims to effectively improve the sample efficiency without compromising the performance. Therefore, we focus on two objectives: whether VDPO can achieve better sample efficiency, and whether VDPO can provide comparable performance with SOTA techniques.
>
> In terms of the **sample efficiency**, our VDPO can successfully hit the threshold in 4 out of the 9 tasks (HumanoidStandup-v4, Pusher-v4, Reacher-v4 and Swimmer-v4) when delays increase to 25 and 50, while the SOTA baseline AD-SAC hit the threshold in 3 out of the 9 tasks while requiring more samples in majority tasks.
>
> In terms of **performance**, when delays are increased to 50, the SOTA baseline AD-SAC achieves the best performance in 5 out of the 9 tasks (Ant-v4, HalfCheetah-v4, Hopper-v4, HumanoidStandup-v4, and Swimmer-v4), and our VDPO achieves the best performance in 3 out of the 9 tasks (Humanoid-v4, Pusher-v4 and Reacher-v4), showing a comparable performance with the SOTA baseline.
> We believe our empirical results based on **common experiment setting and fair comparison** demonstrate that our VDPO achieves better sample efficiency than SOTA baselines and **does provide comparable performance** with the SOTAs.
>
> [1] Liotet, Pierre, et al. "Delayed reinforcement learning by imitation.", 2022.
>
> [2] Kim, Jangwon, et al. "Belief projection-based reinforcement learning for environments with delayed feedback.", 2023.
>
> [3] Wu, Qingyuan, et al. "Boosting Reinforcement Learning with Strongly Delayed Feedback Through Auxiliary Short Delays.", 2024.
>
> [4] Du, Jing, et al. "Sensory manipulation as a countermeasure to robot teleoperation delays: system and evidence.", 2024.
>
> [5] Penin, Luis F. "Teleoperation with time delay-a survey and its use in space robotics.", 2002.
>
> [6] Sheridan, Thomas B. "Space teleoperation through time delay: Review and prognosis.", 1993.
>
> [7] Liotet, Pierre, et al. "Learning a belief representation for delayed reinforcement learning.", 2021.
>
> [8] Xia, Bo, et al. "DEER: A Delay-Resilient Framework for Reinforcement Learning with Variable Delays.", 2024.

---

> ### Comment · Reviewer_FFJG · 2024-08-09
>
> Thanks for the detailed responses, which solve most of my concerns. However, there is at least a soundness issue in your reply to weakness 3. For instance, if the oracle conditional probability distribution $p(s_{t-\Delta+i}\mid x_t)$ is a multimodal distribution (e,g, has two peaks), the mse loss will lead to the predicted $s_{t-\Delta+i}$ falling in the valley between these two peaks. The proposed method may fail in such a situation. Hence, the assumption of Gaussian distribution is necessary in the reviewer's opinion. In that case, I reduce my score of soundness to 1 and hope the authors pay more attention to this problem.

---

> ### Author Response · Authors · 2024-08-09
>
> Thank you very much for the comment. Below, we will further clarify that our approach does not assume that the belief $b$ has to follow a Gaussian distribution in Eq. (10) and our theory is sound regardless of the distribution of $b$.
>
> First, the theoretical results on the sample complexity (Lemma 3.5, line 171) and convergence point (Lemma 3.7, line 181) of VDPO only depend on the ground-truth belief function $b$ and the optimal reference policy $\pi^*$ being given. Therefore, the theory of this work is sound regardless of the specific distribution of the belief.
>
> Moreover, MSE is a commonly used loss function for general random variable, as evidenced in [1] (P283, the MSE is the lose function for Bernoulli variables; P286, the MSE is the lose function for exponential variable). Similar to many existing works, e.g., Eq. (3) in [2] and Eq. (1) in [3], the MSE in Eq. (10) in our implementation can be validly used without prior knowledge of the distribution. The assumption of the random variable following Gaussian distribution is needed only if we want to consider the maximum likelihood estimation, and its impact on our approach is only on practical performance but not soundness. Furthermore, as mentioned in the Limitation and Future Works section (Lines 261-262), we focus on the deterministic cases in this paper, where using MSE does not affect performance.
>
> As mentioned in the Limitation and Future Works section (Lines 262-265), we plan to investigate stochastic applications in future, in which case if the belief $b$ does not follow Gaussian distribution, using MSE loss in Eq. (10) may not achieve the best performance. Note that beyond the distribution of the belief $b$, other factors, such as the performance gap between the reference policy $\pi$ and delayed policy $\pi_\Delta$ [4], may also affect the performance in stochastic environments. It would indeed be interesting to study the impact of these factors on our approach's practical performance and explore options to improve the performance when necessary (i.e., other loss functions).
>
> To sum up, we emphasize that our work is theoretically sound regardless of the distribution of the belief in Eq. (10). Moreover, MSE is a commonly used loss function in practical reinforcement learning [2, 3] without the Gaussian distribution assumption, and this work focuses on deterministic benchmarks, where the belief is a deterministic mapping and MSE is thus the best unbiased estimator. We acknowledge that in stochastic MDPs, using MSE under non-Gaussian belief may lead to suboptimal performance, as pointed out by the reviewer's insightful comment. We plan to investigate the stochastic scenarios and various factors that may affect the practical performance (e.g., the distribution of the belief, the choice of the loss function, and the performance gap between reference policy and delayed policy) in our future work.
>
> [1] Lehmann, Erich L., and George Casella. Theory of point estimation. Springer Science and Business Media, 2006.
>
> [2] De Bruin, Tim, et al. "Integrating state representation learning into deep reinforcement learning." IEEE Robotics and Automation Letters 3.3 (2018): 1394-1401.
>
> [3] Ota, Kei, et al. "Can increasing input dimensionality improve deep reinforcement learning?." International Conference on Machine Learning. PMLR, 2020.
>
> [4] Wu, Qingyuan, et al. "Boosting Reinforcement Learning with Strongly Delayed Feedback Through Auxiliary Short Delays." International Conference on Machine Learning. PMLR, 2024.

---

> ### Comment · Reviewer_FFJG · 2024-08-12
>
> Thanks for the detailed response. So $s_{t-\Delta+i}$ in equation 11 is not obtained from the belief function, but from the ground truth in the delayed-free environment, right?

---

> > ### Author Response · Authors · 2024-08-12
> >
> > Thank you very much for the comment.
> > We clarify that $s_{t-\Delta+i}$ in Eq. (11) is the ground truth state in the delay-free environment. For the sake of distinction with the true state $s$, we use $\hat{s}$ to denote the predicted state from the belief $b$ (Line 203).
> >
> >
> > Eq. (11) is to minimize the behaviour difference between the reference policy $\pi$ and the delayed policy $\pi_\phi^{(i)}$ over the observed pair $x_t$ and $s_{t-\Delta+i}$ for any delay $1\leq i\leq \Delta$ (Line 214), motivated by Eq. (3) in [1]. The belief $b$ in Eq. (10) is only used for obtaining a shared encoder with delayed policy (Line 208).
> >
> > Note that if we only consider the $\Delta$-delayed policy, Eq. (11) can be simplified as
> > $$
> > \nabla_\varphi \text{KL}(\pi^{(\Delta)}_\varphi(\cdot|x_t)||\pi(\cdot|s_t)).
> > $$
> >
> > We propose the general form Eq. (11) in the paper to accommodate the stochastic delays within $[1,\Delta]$.
> >
> > [1] Liotet, Pierre, et al. "Delayed reinforcement learning by imitation." International Conference on Machine Learning. PMLR, 2022.

---

> > > ### Comment · Reviewer_FFJG · 2024-08-13
> > >
> > > Thanks for your responses. The reviewer misunderstands the usage of the belief function. In that case, the formulation is correct and the reviewer will raise the score to 6.

---

> > > > ### Author Response · Authors · 2024-08-13
> > > >
> > > > Sincerely appreciate raising your rating and your time and effort in reviewing and giving insightful comments on our manuscript.

---

### Author Rebuttal · Authors · 2024-08-05

# General Response
We sincerely appreciate the insightful comments and feedback from all the reviewers.
The main contribution of this work is to address the sample efficiency issue in delayed RL by first introducing variational inference to reformulate the original problem and then solving the high-dimensional policy learning based on behaviour cloning. We are pleased that our approach, VDPO can effectively **improve the sample complexity without compromising the performance**, supported by both theoretical analysis and experimental results.

Regarding the main concern on inference time from Reviewer FFJG, we clarify that existing delayed RL works [1, 2, 3], including this paper, all use **sample efficiency and performance** as the evaluation metrics. The reason for not considering inference time is that commonly adopted neural architectures, e.g., MLP, transformer, etc., for policy representation share similar inference times, which are all significantly smaller (at least three orders of magnitude less) than one normal control cycle. Such inference times can be safely neglected considering the typical delay (multiple control cycles) in real-world applications. The detailed comparison between MLP inference time, transformer inference time, and the real-world delays can be found in answer to Weakness 2 by Reviewer FFJG.
We appreciate Reviewer FFJG's comment and will add related discussion in the revised version to avoid any potential confusion.


[1] Liotet, Pierre, et al. "Delayed reinforcement learning by imitation." International Conference on Machine Learning. PMLR, 2022.

[2] Kim, Jangwon, et al. "Belief projection-based reinforcement learning for environments with delayed feedback." Advances in Neural Information Processing Systems 36 (2023): 678-696.

[3] Wu, Qingyuan, et al. "Boosting Reinforcement Learning with Strongly Delayed Feedback Through Auxiliary Short Delays." Forty-first International Conference on Machine Learning. PMLR, 2024.

---

### Decision · Program_Chairs · 2024-09-25

**Decision:**

Accept (spotlight)

**Comment:**

This paper tackles the challenging problem of delayed observation and proposes Variational Delayed Policy Optimization (VDPO), which reformulates delayed RL as a variational inference problem. I think the idea is novel, and it receives the unanimously acceptance by the reviewers. The impressive part is that VDPO is able to improve the sample efficiency significantly with 50% less amount of samples and this is strong. I recommend the paper be accepted. I also recommend the authors to consider how to expand the existing Mujoco experiments to more realistic settings such as real robotics.